# High NaCl Concentrations in Water Are Associated with Developmental Abnormalities and Altered Gene Expression in Zebrafish

**DOI:** 10.3390/ijms25074104

**Published:** 2024-04-07

**Authors:** Denis A. Seli, Andrew Prendergast, Yagmur Ergun, Antariksh Tyagi, Hugh S. Taylor

**Affiliations:** 1Department of Obstetrics, Gynecology and Reproductive Sciences, Yale School of Medicine, New Haven, CT 06520, USA; 2Yale Zebrafish Phenotyping Core, Yale School of Medicine, New Haven, CT 06520, USA; 3IVIRMA Global Research Alliance, IVIRMA New Jersey, Marlton, NJ 08053, USA; 4Department of Genetics, Yale School of Medicine, New Haven, CT 06520, USA

**Keywords:** environment, NaCl, zebrafish, embryo, swim bladder development, sonic hedgehog pathway

## Abstract

Salt is frequently introduced in ecosystems, where it acts as a pollutant. This study examined how changes in salinity affect the survival and development of zebrafish from the two-cell to the blastocyst stage and from the blastocyst to the larval stage. Control zebrafish embryos were cultured in E3 medium containing 5 mM Sodium Chloride (NaCl), 0.17 mM Potassium Chloride (KCL), 0.33 mM Calcium Chloride (CaCl_2_), and 0.33 mM Magnesium Sulfade (MgSO_4_). Experiments were conducted using increasing concentrations of each individual salt at 5×, 10×, 50×, and 100× the concentration found in E3 medium. KCL, CaCl_2_, and MgSO_4_ did not result in lethal abnormalities and did not affect early embryo growth at any of the concentrations tested. Concentrations of 50× and 100× NaCl caused embryonic death in both stages of development. Concentrations of 5× and 10× NaCl resulted in uninflated swim bladders in 12% and 65% of larvae, compared to 4.2% of controls, and caused 1654 and 2628 genes to be differentially expressed in blastocysts, respectively. The ATM signaling pathway was affected, and the Sonic Hedgehog pathway genes Shh and Ptc1 implicated in swim bladder development were downregulated. Our findings suggest that increased NaCl concentrations may alter gene expression and cause developmental abnormalities in animals found in affected ecosystems.

## 1. Introduction

Salt is one of the many types of pollutants introduced into ecosystems, and it can harm organisms. There are many reasons unnatural levels of salt can be found in the environment, including due to pollution from human activity such as the spread of salt on roads to melt ice in the wintertime [1,2]. Salt concentrations can also change significantly due to natural factors, like changes in river flow and precipitation [3].

Heightened salt concentrations can have various effects on organisms who have not adapted to saline environments. Multiple studies have demonstrated that changes in salinity can affect plants. In one study, Wilson et al. (2019) grew freshwater sawgrass in heightened salinity and found that in the short term plants and ecosystems initially grew significantly better in heightened salinity than in their natural environment. However, when exposed to this salt for an extended period, they experienced stress, which resulted in less stable soil and root structures [4]. Another study investigated the impact of high salinity concentrations on mangrove trees and found that they significantly affected physical characteristics of the trees, such as seed germination traits and fruit weight [3]. Overall, the impact of salinity on plant growth and survival is well investigated, and heightened levels often have a significant effect on plant life, often negative.

Changes in salinity can also have effects on animal life. One study demonstrated that changes in salinity can have a significant impact on the growth and reproduction of insects such as the Collembola [5]. Similarly, the impacts of salinity on vertebrates can also be significant and detrimental. Elevated salinity levels in freshwater environments can disrupt the osmoregulatory mechanisms of freshwater fish, leading to physiological stress and potentially adverse effects on their health and survival. Specifically, the changes in the osmotic gradient between their bodies and the surrounding water may lead to dehydration or excess water uptake. This stress can impact various physiological processes, including metabolism, hormone regulation, and immune function [6,7]. Salt can play a role in how animals grow and survive, but there are still few studies on vertebrates.

The zebrafish is a freshwater fish naturally found throughout Central Asia [8]. This fish has been adopted as a model by many because of its multiple scientific advantages, such as transparent embryos, external development, easy growth and manipulability, and possession of genes relevant to mammal and human work [9]. Specifically, the zebrafish is a vertebrate and has early developmental processes (e.g., gastrulation, neurulation, and organogenesis) similar to humans, making it a relevant developmental model [10].

In this study, we examined how changes in the salinity concentration would affect the survival and development of zebrafish. By examining zebrafish development, including a dose response to commonly used salts (NaCl, KCl, CaCl_2_, and MgSO_4_), we determined if these salts were detrimental to development. Two stages of embryo development, two-cell to blastocyst and blastocyst to larval, were examined for developmental impact. Survival rates and developmental parameters were measured. Furthermore, we performed molecular analyses to determine if the increased NaCl concentrations affected gene expression in two-cell zebrafish embryos cultured to the blastocyst stage.

## 2. Results

### 2.1. Impact of Increased Salt Concentrations on Zebrafish Development from the Blastocyst to Larval Stage

Data were gathered on how well zebrafish were able to survive in different concentrations of different salts from the blastocyst to larval stage (see Appendix A for treatments and age groups). As shown in Figure 1, the treatments with CaCl_2_ and MgCl_2_ had no effect on survival (*p* = 0.9 and *p* = 0.8, respectively) (Figure 1F,G), even when the concentration was as high as 100× the natural concentration. When the zebrafish were treated with KCl at higher concentrations such as 50× and 100×, late survival was substantially affected (*p* < 0.05) (Figure 1E). However, this is likely due to the fact that high concentrations of KCl impair cardiac conductivity, leading to arrhythmia and cardiac arrest; the zebrafish in fact does not require active circulation for gas exchange until later in development [11]. The most drastic effect on survival was seen with NaCl, where at 50× and 100× the natural concentration, zebrafish did not survive, whereas survival was not affected at 5× or 10× (*p* < 2 × 10^−16^) (Figure 1D).

In addition to survival, zebrafish were assessed for the presence of gross morphologic abnormalities, including defects in axis formation, craniofacial anatomy, and somitogenesis, as well as for proper formation of the eyes, nervous system, otic vesicle, heart, pigmentation, and gastrointestinal tract (Figure 1). As shown in Figure 1A, at the standard E3 concentration, morphology is normal, with the swim bladder inflated and normal cardiac anatomy. However, zebrafish grown in 5× and 10× NaCl developed uninflated swim bladders (Figure 1B). This was especially common in the 10× NaCl treatment (Figure 1H), which was the highest treatment of NaCl that maintained survival. Treatments using increasing MgCl_2_, CaCl_2_, and KCl had little impact on the proportion of uninflated swim bladders found in the population (Figure 1I–K). Finally, while other salts did not seem to cause significant morphologic abnormalities in any of the concentrations tested, cardiac arrest led to substantial blood pooling in the heart in some fish grown at higher concentrations of KCl (Figure 1C).

### 2.2. Impact of Increased Salt Concentrations on Zebrafish Development from the Two-Cell to Blastocyst Stage

The survival of zebrafish in different salt concentrations was also tested from the two-cell to blastocyst stage. As shown in Figure 2, NaCl had no effect on survival at concentrations 5× and 10× the natural concentration (Figure 2K). However, NaCl caused death at 50× and 100× the natural concentration (*p* = 6 × 10^−10^). KCl, CaCl_2_, and MgCl_2_ had no effect on survival at concentrations 5×, 10×, 50×, and 100× the natural concentration (*p* = 1) (Figure 2L–N). Representative images of embryo morphologies are shown in Figure 2A–J.

### 2.3. Altered Expression of Sonic Hedgehog Pathway Genes in Zebrafish Embryos Treated with Higher NaCl Concentrations

The first two sets of experiments described above revealed a number of findings regarding the effect of NaCl on zebrafish development. First, we found that 50× and 100× concentrations of NaCl caused the demise of zebrafish in both developmental stages (two-cell to blastocyst and blastocyst to larvae). We also observed that 5× and 10× NaCl caused a failure to develop a swim bladder in the larval stage (Figure 1), whereas there were no morphologic changes in blastocysts (Figure 2).

Therefore, we next investigated whether there were any molecular changes that occurred during blastocyst development that could explain the abnormal swim bladder phenotype in zebrafish larvae treated with 5× and 10× concentrations of NaCl. The hedgehog signaling pathway plays an important role in organ development in zebrafish, mice, and humans [12]. This pathway is known to contribute to the development of the swim bladder in zebrafish [13]. We therefore decided to examine whether increased NaCl alters the expression of Sonic Hedgehog (Shh) and Patched 1 (Ptc1), two genes that are an essential part of the hedgehog signaling pathway and are associated with zebrafish swim bladder development [14].

qRT-PCR was used to test zebrafish embryos from the NaCl treatments E3 (baseline), 5×, and 10× to see if there was a significant difference in the expression of Shh and Ptc1. We found that compared to the control samples, fish grown in treatments of 5× and 10× natural NaCl concentrations had significantly lower expression of Shh (*p* < 0.0001) and Ptc1 (*p* < 0.0001) (Figure 3).

### 2.4. Transcriptomic Landscape of Zebrafish Embryos Treated with Higher NaCl Concentrations

After finding that the hedgehog signaling pathway genes Shh and Ptc1 are suppressed in zebrafish blastocysts grown in 5× and 10× NaCl concentrations, we performed RNA seq analyses to further characterize the impact of these treatments on overall gene expression in zebrafish blastocysts.

Hierarchal clustering of the differentially expressed genes partitioned into distinct clusters showed differential gene expression in zebrafish blastocysts grown in 5× and 10× NaCl compared to the baseline culture condition (E3) (Figure 4). There were a total of 1654 differentially expressed genes in the 5× NaCl group compared to baseline (padj < 0.05) (Figure 4A,B), 1030 downregulated and 624 upregulated (Figure 4A,B). In the 10× NaCl group, there were 2628 genes that were differentially regulated compared to baseline (padj < 0.05) (Figure 4D,E); 1491 were downregulated and 1137 upregulated (Figure 4D,E).

A functional enrichment analysis was performed to determine Gene Ontology (GO), and it revealed that differentially expressed genes in the 5× NaCl group were mainly in the spliceosome cycle, ATM signaling (DNA damage response), sirtuin signaling, and kinetochore metaphase signaling (Figure 4C). Similarly, the 10× NaCl group affected ATM signaling, kinetochore metaphase signaling, and microRNA biogenesis (Figure 4F). When Physiological System Development and Function was assessed, both concentrations affected embryo development and organismal development (Figure 4C,F).

## 3. Discussion

In this study, we found numerous negative impacts on zebrafish that were associated with increased salt concentrations in the environment. Two developmental stages were tested, from two-cell to blastocyst and from blastocyst to larvae, and the most prominent effects were observed with NaCl. In both stages, zebrafish did not survive the highest concentrations of NaCl tested (50× and 100×). Moderately elevated NaCl concentrations (5× and 10×) were associated with a decreased incidence of swim bladder inflation, but they did not cause morphologic changes during early zebrafish embryo development (two-cell to blastocyst). Despite the absence of visible abnormalities in blastocysts cultured in 5× and 10× NaCl, we found that these embryos have suppressed hedgehog pathway gene expression (Shh and Ptc1) and display large-scale changes in overall gene expression, affecting multiple pathways related to development.

Road salts, typically used for de-icing and snow removal on roads during winter, can have various harmful effects on ecosystems. Elevated salinity levels can alter the chemical composition of water, affecting pH levels and disrupting the balance of ions essential for aquatic life. Wildlife species dependent on freshwater habitats may be negatively affected by road salts [15]. Amphibians, reptiles, birds, and mammals may experience adverse health effects or habitat loss due to changes in water quality and vegetation composition [16]. The widespread use of road salts can disrupt natural ecosystems and create ecological imbalances. Species adapted to low salinity levels may be outcompeted by salt-tolerant species, leading to shifts in community composition and loss of biodiversity. Different salt types are being used for de-icing, and the choice of salt is often determined by cost, effect on concrete, and effectiveness in each temperature. Our findings suggest varying harmful effects caused by these salts, potentially related to cellular transport mechanisms and specificity of effects on molecular pathways.

Zebrafish are a commonly used model to study biological mechanisms and human disease for a multitude of reasons, including their fully sequenced genome, easy genetic manipulation, high fecundity, transparent embryos, external fertilization, and rapid development. Furthermore, zebrafish have much in common with humans as well as other organisms used as models, such as mice, including similar embryonic and postembryonic development and conserved genetic pathways. Because of these similarities, not only can zebrafish research be generalized to similar types of fish, but it can also be extrapolated to other vertebrates. In the current study, using the zebrafish model, we were able to test the impact of several salts and varying concentrations on different stages of development in a relatively short period of time.

The hedgehog pathway is a key regulator of cell differentiation and proliferation and plays an important role in organ development in animal models and human [17,18,19]. In this study, uninflated swim bladders were observed in zebrafish grown in 5× and 10× NaCl concentrations. An uninflated swim bladder is a serious developmental abnormality, as the gas-filled swim bladder is necessary in providing the lift that allows fish to attain neutral buoyancy. We therefore decided to investigate the molecular mechanisms that may be responsible for this morphological deformity, and because the hedgehog pathway is heavily involved in the development of the swim bladder in zebrafish, we examined whether the expression of genes in this pathway were affected [14,20]. We found hedgehog pathway genes Shh and Ptc1 to have lower expression in the zebrafish blastocysts grown in 5× and 10× NaCl. The altered expression of hedgehog pathway genes may have implications on zebrafish development beyond the swim bladder, as this pathway is linked to the development of many organs and physiological systems [21]. For this reason, pollutants that negatively impact the hedgehog pathway and other crucial pathways could be harmful in unforeseen ways, for zebrafish and humans alike.

To further investigate the molecular impacts of the increase in concentration of NaCl, RNA sequencing analyses were performed. The 5× and 10× NaCl groups had 1654 and 2628 differentially expressed genes compared to controls, respectively. When we compared the 5× to the 10× NaCl treatment, we only found 28 genes to be differentially expressed, suggesting the effect of NaCl treatment to be specific, only to be amplified with a higher (10×) concentration. This conclusion was further supported by the detection of similar pathways that were affected by the 5× and 10× treatments: ATM signaling, the cell cycle, cell death and survival, and embryologic and organismal development. The identification of ATM signaling as a shared pathway affected by both 5× and 10× NaCl concentrations is noteworthy. ATM, the gene mutated in the disorder ataxia-telangiectasia, is a protein kinase that is a central mediator of responses to DNA double-strand breaks in cells [22]. ATM signaling plays a key response to DNA damage in response to oxidative stress [23], and our findings suggest that increased salt may induce DNA strand breaks and result in cell death. Overall, these results demonstrate the wide variety of changes than can result from a small change in external circumstances such as heightened salinity, culminating in abnormal organ development in zebrafish.

With the data from this study, we can continue to investigate how salt is detrimental to ecosystems and organisms as a roadside pollutant. Prior research has clearly demonstrated how harmful increased levels of salinity can be for plant life [24]. The impacts of increased salinity on animals are still contested, with many studies showing negative impacts but with varying degrees, and with different conclusions on the relative harmfulness of different types of salt [25]. The reported impacts on increased salinity range from death to decreased size and altered behavior [25,26]. These findings are still being investigated, and there is currently no agreed-upon model.

Salinity stress in fish and other animals refers to the physiological challenges and adverse effects caused by exposure to elevated salt concentrations in their environment. While some species, such as marine organisms, have evolved mechanisms to tolerate and thrive in saline conditions, others, particularly freshwater species, are more sensitive to changes in salinity levels. Exposure to salinity stress can lead to physiological disruptions, including alterations in ion balance, dehydration, and changes in metabolic rates [27,28]. These stressors can weaken the immune system, impair reproductive functions, and increase susceptibility to diseases. Salinity stress can also have cascading effects on aquatic ecosystems. Changes in salinity levels can disrupt food webs, alter species composition, and affect ecosystem dynamics. For example, fluctuations in salinity can impact the distribution and abundance of aquatic plants, invertebrates, and fish populations. Understanding the ecological implications of salinity stress is crucial for conservation efforts and ecosystem management. Salinity stress poses challenges for fish and other aquatic animals, particularly those adapted to specific salinity ranges [29]. Mitigating the impacts of salinity stress requires comprehensive management strategies that consider both natural variability and human-induced disturbances in aquatic environments.

This study has several limitations. These include the observations being limited to the zebrafish model, lack of data on the long-term effects of salt exposure, and the influence of other environmental factors. Environmental changes often have cumulative and persistent effects that may not be immediately apparent. Focusing only on short-term impacts may overlook delayed or indirect effects on ecosystems and organisms. Ecosystems are composed of diverse species that interact with each other and their environment. Focusing on a single organism may not capture the broader ecological implications of environmental stressors, including effects on other species and ecosystem-level responses.

Although increased salinity can be harmful for vertebrates, further research is still needed to clarify these findings. Salinity has been shown to harm vertebrates in many ways, but the mechanisms for most of these interactions are still unknown. Furthermore, the significance of the type of salt when looking at harmful interactions is still not agreed upon. Research is required to compare the impacts of different types of salt across species to start to determine what attributes (physical or genetic) are associated with what kind of reaction to a type of salt. These studies must be performed in conjunction with molecular analyses to determine if the mechanism that causes harm to these vertebrates is conserved across species, and can be conducted in aquatic vertebrates, such as fish, or mammals like mice that may share the impacted gene pathways. The findings from our study as well as those that follow should guide policy on the concentrations and types of salts that are used on roads.

## 4. Materials and Methods

### 4.1. Zebrafish Culture

Wild-type zebrafish (*Danio rerio*) were used for experiments that were conducted at Yale Zebrafish Research Core, CT, USA. Adult *AB animals were kept on a standard 14/10 h light cycle in 3.0 L tanks with continuous water flow (Iwaki Aquatic, Holliston, MA, USA). Fish were maintained on a diet consisting of Gemma Micro 500 2×/day and cultured brine shrimp 1×/day (Skretting, Stavanger, Norway). Embryos were obtained by natural mating and cultured in E3 medium until the initiation of salt-stress experiments, at which point media were modified as described below. Embryos were maintained at 28.5 °C throughout. Larval euthanasia was carried out by immersion in 0.16 mg/mL MS-222 (Syndel, Ferndale, WA, USA) and subsequent rapid chilling. Animal care was modeled after methods developed at Boston Children’s Hospital [30,31]. All animal care and handling were performed under the approval of Yale IACUC, protocol #2019-20274. Per this protocol, humane endpoints include aberrant swimming or abnormal sinking or floating. Zebrafish larvae were euthanized as described above at the end of the study period or immediately if meeting any of these endpoint criteria. Larvae were monitored every other day. Generally, animals did not die before meeting these criteria.

The impact of environmental salt concentration on zebrafish development was tested in two stages: (1) the blastocyst (2.5 h after fertilization—when the embryo has 256 cells) to larval (6-day-old freely swimming and hunting fish) stage and (2) the 2-cell (0.5 h after fertilization) to blastocyst stage.

The fish were grown at these time intervals in different salinity concentrations. The solutions consisted of 1 of 4 salts (NaCl, KCl, MgCl_2_ and CaCl_2_), and in addition to the baseline concentration of each salt (E3), 4 additional concentrations that were multiples of the baseline were used (5× baseline concentration, 10×, 50×, and 100×) (Appendix A). The natural concentrations of the salts as well as all the test groups are listed in Appendix A. All breeding was conducted at 28.5 °C. Each experiment was repeated three times, and 30–50 embryos or fish were assessed for each salt concentration and stage of development. As the fish grew over the course of their assigned time intervals, data were recorded on survival and anatomic abnormalities. An Olympus MVX10 microscope (Shinjuku, Tokyo, Japan) equipped with 1.0× and 2.0× apochromatic objectives, GFP/mCherry filter sets, and an Olympus DP74 color CMOS camera was used for zebrafish experiments.

### 4.2. Quantitative Reverse-Transcription Polymerase Chain Reaction (qRT-PCR)

RNA was extracted from 10 zebrafish blastocysts per sample using the RNAqueous Microkit according to the instructions of the manufacturer (Thermo Fisher Scientific, Waltham, MA, USA, Cat#AM1931). Each experiment was repeated 3 times, using 3 samples per group in each experiment. cDNA was synthesized using the High-Capacity cDNA Synthesis Kit (Thermo Fisher Scientific, Cat#4368814) at 42 °C for 1 h in a Bio-Rad thermal cycler (Bio-Rad laboratories, Hercules, CA, USA). After this, the concentration of cDNA in each sample was measured using the Qubit dsDNA High Sensitivity Assay Kit (Thermo Fisher Scientific, Cat#Q32851) and diluted to normalize the concentrations to 2 ng/µL. Using this cDNA, a real-time PCR reaction was set up with 13 µL of SYBR green master mix (Applied Biosystems, ABI, Foster City, CA, USA, Cat#4309155), 1 µL of each primer (forward and reverse) for the gene, 2.5 µL of cDNA, and 2.5 µL of water, for a total volume of 20 µL per well. The PCR was run on a Bio-Rad thermal cycler according to the instructions of the manufacturer (Bio-Rad laboratories) for 40 cycles. Initial denaturation was performed at 95 °C for 3 min followed by amplification at 60 °C for 30 s and final extension at 72 °C for 5 min as per protocol. The 2^−∆∆CT^ (cycle threshold) method was used to calculate relative expression levels after normalization to β-actin levels. The primers used for real-time PCR reactions are given in Appendix A.

### 4.3. Statistical Analyses

All statistical tests for the PCR analyses were performed using Graph Pad Prism software (Version 9.4.0), and significance was assessed at *p* < 0.05. Values were analyzed either by Student’s *t* test, One-way ANOVA, or Two-way ANOVA as described in each figure legend. Survival is presented as standard Kaplan–Meier plots, and statistical significance was assessed using a log-rank survival test; all survival data were processed using R (Version 4.2.3, 15 March 2023) and the ‘survival’ and ‘survminer’ libraries.

### 4.4. RNA Extraction and Sequencing

RNA was extracted from zebrafish blastocysts (10 for each sample) and amplified and prepped using the NEBNext Single Cell/Low Input RNA Library Prep Kit for Illumina (Cat#E6420S, Illumina, San Diego, CA, USA) for sequencing following the manufacturer’s protocol. cDNA was amplified and measured by qRT-PCR using a commercially available kit (Roch KAPA Biosystems Cat#KK4854, Wilminghton, MA, USA), and insert size distribution was determined by the Agilent Bioanalyzer. Indexed libraries were then pooled and sequenced on Illumina’s NovaSeq6000 platform with 101 bp paired-end reads. In total, 15 samples (5 samples per experimental group) were analyzed, and approximately fifty million paired-end reads were achieved for each sample.

### 4.5. RNA Sequencing

RNA sequencing was performed at the Yale Center for Genome Analysis at Yale University, USA. Image analysis, base calling, and generation of sequence reads were produced using the HiSeq Control Software v2.0 (NCS) and Real-Time Analysis Software (NVivo, Version 4.0) (RTA). Low-quality reads were trimmed, and adaptor contamination was removed using Trim Galore (v0.5.0). Trimmed reads were mapped to the Zebrafish Build 11 (GRCz11) using HISAT2 (v2.1.0) [32]. Gene expression levels were quantified using StringTie (v1.3.3b) with gene models (v27) from the GENCODE project [33]. Differentially expressed genes were identified using DESeq2 (v 1.22.1) [34]. If the adjusted *p*-value was less than or equal to 0.05, the genes were identified as differentially expressed. A matrix of the Pearson correlation coefficient was created using R Package.

Ingenuity Pathway Analysis (IPA) Ingenuity Systems QIAGEN (Content version: 76765844, 2022, www.qiagen.com/ingenuity (accessed on 8 October 2023), Redwood City, CA, USA) software was used to perform pathways analyses. A false discovery rate (FDR) ≤ 0.05 was assigned as differentially expressed in different comparisons. Each gene symbol was mapped to its corresponding gene object in the Ingenuity Pathways Knowledge Base. Expression pattern clusters were generated by the unsupervised hierarchical clustering analysis and the K-means clustering algorithm using R. DAVID and IPA were used to reveal the Gene Ontology (GO) and pathways.

## Figures and Tables

**Figure 1 ijms-25-04104-f001:**
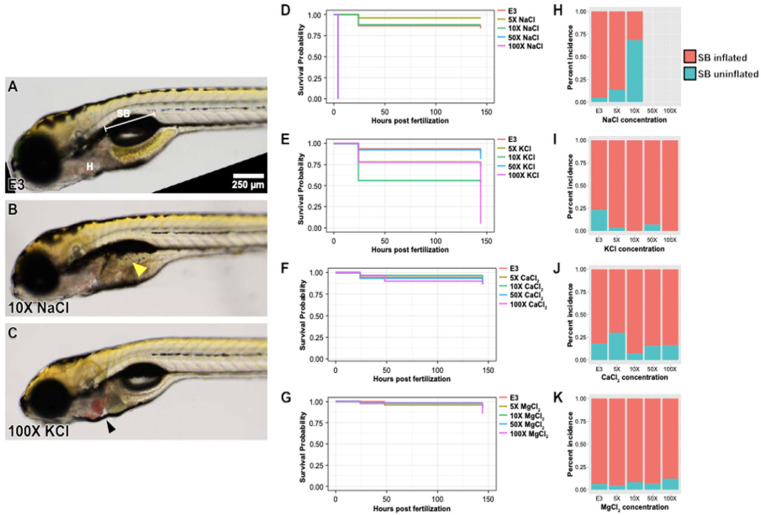
Elevated concentrations of four road salt components generate minor developmental phenotypes. (**A**) A 6 dpf zebrafish larvae maintained in standard E3 medium. Position of inflated swim bladder (SB) and normally functioning heart (H) are indicated. (**B**) Same stage larvae maintained at 10× (50 mM) the normal concentration (5 mM) of NaCl. Note that morphology is generally normal, but swim bladder is not inflated (yellow arrowhead). (**C**) Same stage larvae maintained at 100× (16 mM) the normal concentration (1.6 mM) of KCl. Note that blood is pooling abnormally in the heart (black arrowhead) due to impaired cardiac function. Swim bladder inflation is, however, normal. (**D**) Survival curve of all 4 tested NaCl concentrations. The 50× and 100× incubations are not survivable; embryo dies within first hour of osmotic stress. Otherwise, survival is generally fine through 6 days. (**E**) Similar survival curve, but for KCl. Embryos at higher concentrations of KCl eventually die, likely due to lack of cardiac function (this function is unnecessary at earlier stages but becomes progressively more important as the larva ages). (**F**) Similar survival curve, but for CaCl_2_. Survival is unimpaired across all concentrations. (**G**) Similar survival curve, but for MgCl_2_. Survival is unimpaired across all concentrations. (**H**) Incidence of swim bladder inflation in NaCl-treated larvae. At 10× NaCl concentration, swim bladder fails to inflate ~70% of the time. We do not observe anything comparable for any of the other salt components (**I**–**K**).

**Figure 2 ijms-25-04104-f002:**
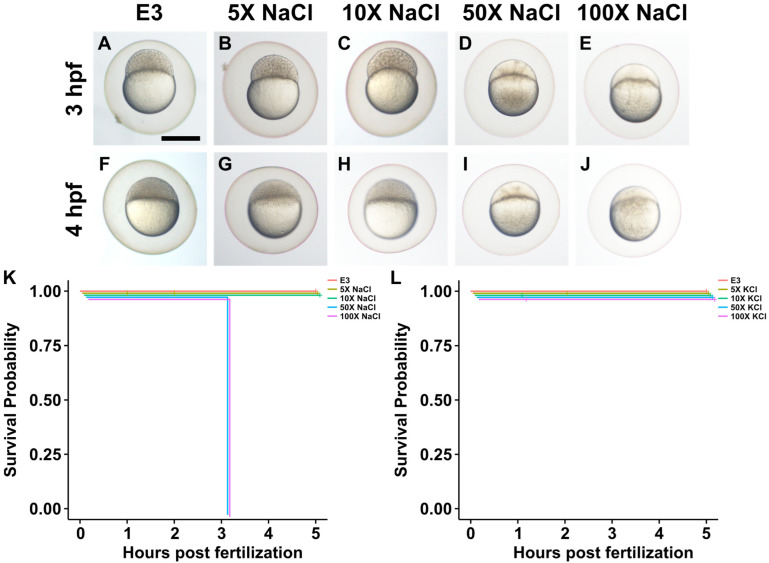
Detailed survival analysis of all four road salt components through late blastula stages. (**A**) A 3 hpf (although closer to 256-cell stage) zebrafish embryo maintained in standard E3 medium. Scale bar: 500 µm. (**B**) Same stage embryo maintained at 5× (25 mM) the normal concentration (5 mM) of NaCl. (**C**) Same stage embryo maintained in 10× (50 mM) NaCl. (**D**) Same stage embryo maintained in 50× (250 mM) NaCl. Note that at this concentration, cell boundaries are not evident; embryo is no longer dividing and in fact the earlier (2/4/8) cell divisions have apparently reversed. (**E**) Same stage embryo maintained in 100× (500 mM) NaCl. Same result as in (**D**) is evident. (**F**–**J**) Several 4 hpf (although closer to oblong stage) embryos treated with same NaCl concentrations as in (**A**–**E**) exhibiting similar results. (**K**) Survival curve for all NaCl concentrations, evaluated hourly for first 5 h of development. All embryos for 50× and 100× NaCl concentrations die (as defined here by failure to maintain cell-cell boundaries and continue dividing) by 3 hpf. (**L**–**N**) Survival curves for other road salt components do not affect survival through later blastocyst stages.

**Figure 3 ijms-25-04104-f003:**
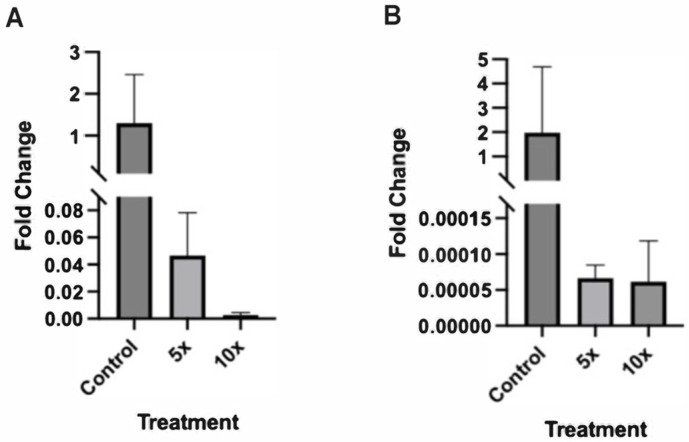
Fold changes (2^−∆∆CT^) of Shh (**A**) and Ptc1 (**B**) across 3 treatments of NaCl concentration (E3, 5×, and 10×). For both genes, gene expression in both 5× and 10× natural salinity were significantly lower than expression in control.

**Figure 4 ijms-25-04104-f004:**
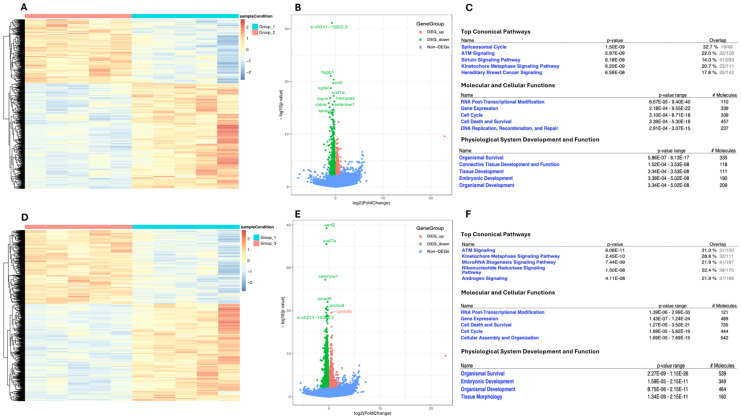
Gene expression is altered in zebrafish raised in heightened NaCl concentrations. (**A**,**D**) Heatmap illustration showing genes differentially expressed between zebrafish raised in 5× and 10× NaCl concentrations and E3. The color spectrum indicates normalized levels of gene expression, with red representing high and blue representing low. (**B**,**E**) Volcano plots for RNA seq of zebrafish comparing those raised in 5× and 10× NaCl to those raised in E3. Red spot represents upregulation, green spot represents downregulation. (**C**,**F**) Table showing noteworthy genetic change from zebrafish grown in E3 to those grown in 5× and 10× concentrations of NaCl. #: number of molecules.

## Data Availability

Data will be made available to investigators upon request.

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
