# Peer review of "High NaCl Concentrations in Water Are Associated with Developmental Abnormalities and Altered Gene Expression in Zebrafish"

_ijms, 2024, doi:10.3390/ijms25074104_

Round 1
Reviewer 1 Report
Comments and Suggestions for Authors
comments are attached in review report

Comments on the Quality of English LanguageThe MS is recommended for a through revision in terms of gramatical mistakes, phrases.
Reviewer 2 Report
Comments and Suggestions for Authors
The authors investigated the role of different salts in zebrafish embryo development with particular attention to NaCl.
The rational is interesting but quite faint since the amount of road salt as pollutants in sweetwater lakes should be trascurable. Did you have data about ambient damage or problems in some areas ? please extend introduction and discussion about the rational.
Line 67 – “commonly used salt” whereas Line 98 – “four road salts” . I don’t understand what has been used for the experiments. Road salt is NaCl but may contain other impurity and it is largely commercial. Laboratory NaCl is pure. Did you use pure NaCl or not ? and what about other salts Ca and Mg ?
Figure 1 – in the left panel, should be better include also KCl treated embryo
Legend figure 3 – please rephrase avoiding “bar graphs “ expression.
Line 316 – underlined text ? RNA sequencing ? please revise 4.4 and 4.5 in metherial&methods
Reviewer 3 Report
Comments and Suggestions for Authors
The abstract should be a total of about 200 words maximum.
Line 45: Wilson et al. add date
Line 56: explain the effect
Line 58: if present, add studies on mammals and fishes or write that in literature are not present
Line 75: insert table 1
Line 85: insert here figure 1
Line 88: add references
Line 88: I suppose it is fig1 and not fig 2
Line 90: add in the main text the reference to fig 1 D-E-F-G
Fig 1 b: add an arrow showing uninflated swim bladders
Line 117: add in the main text the reference to fig 2 A to J
Line 121: insert here figure 2
Line 292: add information about microscope and program used tp make photo of zebrafish
Line 295: add catalogue number
Line 298: add catalogue number
Line 302: add catalogue number
Line 305: add the protocol of PCR
Suppl table 2: insert the amplicone size
Line 320: add catalogue number
Write reference according instruction https://www.mdpi.com/journal/ijms/instructions
Comments on the Quality of English Language
Minor editing of English language required
Round 2
Reviewer 1 Report
Comments and Suggestions for Authors
now article can be accepted
Author Response
We'd like to thank to reviewer for considering our manuscript for publication.